# Self-selected interval judgments compared to point judgments: A weight judgment experiment in the presence of the size-weight illusion

**Nichel Gonzalez**[1]*, **Ola Svenson**[1,2], **Magnus Ekström**[3,4], **Bengt Kriström**[5], **Mats E. Nilsson**[1]

**1** Department of Psychology, Gösta Ekman Laboratory, Stockholm University, Stockholm, Sweden, **2** Decision Research, Eugene, Oregon, United States of America, **3** Department of Forest Resource Management, Swedish University of Agricultural Sciences, Umeå, Sweden, **4** Department of Statistics, USBE, Umeå University, Umeå, Sweden, **5** Department of Forest Economics, Swedish University of Agricultural Sciences and CERE, Umeå, Sweden

* nichel.gonzalez@psychology.su.se

**Data Availability Statement:** All relevant data are available on Figshare (DOI: 10.17045/sthlmuni. 19213245).

## Abstract

Measurements of human attitudes and perceptions have traditionally used numerical point judgments. In the present study, we compared conventional point estimates of weight with an interval judgment method. Participants were allowed to make step by step judgments, successively converging towards their best estimate. Participants estimated, in grams, the weight of differently sized boxes, estimates thus susceptible to the size-weight illusion. The illusion makes the smaller of two objects of the same weight, differing only in size, to be perceived as heavier. The self-selected interval method entails participants judging a highest and lowest reasonable value for the true weight. This is followed by a splitting procedure, consecutive choices of selecting the upper or lower half of the interval the individual estimates most likely to include the true value. Compared to point estimates, interval midpoints showed less variability and reduced the size-weight illusion, but only to a limited extent. Accuracy improvements from the interval method were limited, but the between participant variation suggests that the method has merit.

## Introduction

Most measurements include some degree of error, but often it is possible to find an interval that most likely includes the true value. To illustrate, the speed of light has been measured since 1870. Initially, these measurements were given as rather wide intervals, illustrating the uncertainty involved. The intervals became successively smaller and converged towards the current speed of light estimate [1]. Similar to natural science measurement, human judgments also include varying degrees of uncertainty. In the present study, we used an interval judgment method where participants were allowed to make step by step judgments, successively converging towards their best estimate. Hence, the interval judgment method used in this study is

**Funding:** Marianne \& Marcus Wallenberg Foundation, Project MMW 2017.0075 \emph{The right question ? new ways to elicit quantitative information in surveys}. Recipient: BK (main applicant, ME och MN subapplicants) web: https://mmw.wallenberg.org/startsida No commercial companies supported the study "The funders had no role in study design, data collection and analysis, decision to publish, or preparation of the manuscript."

**Competing interests:** The authors have declared that no competing interests exist.

conceptually similar to the one describing the physicists' measurements of speed of light which became more and more exact over the years.

We used an interval judgment method, called self-selected intervals [2, 3], and compared the result with traditional point judgments. The judgment task used for comparisons of the judgment methods was a weight judgment task. The judged weights were boxes of different sizes, thereby creating the size-weight illusion (SWI). To clarify, SWI refers to the fact that if two similar objects have the same weight but are of different size, the smaller of the two will be perceived as heavier, and this illusion is a very robust phenomenon [4]. The inclusion of SWI in our experiment allowed us to test, not only the accuracy of judgments, but also if the interval method reduces the illusion. Furthermore, including the SWI adds an element of perceptual variability within individuals, and we explored if the interval method reduces this kind of variability. Traditionally, researchers have tried to avoid the stimulus-error [5], by for example using magnitude estimation. However, this study aims to find out if the interval method can improve accuracy of judgments, in addition to reducing the SWI as well as obtaining more stable estimates. Hence, we have measured a judgment of the true state of the world (objective weight), and such judgments include prior information/knowledge of weight. This means that the accuracy of judgments in this study includes deliberate cognition, while the SWI occurs involuntary cognition on a perceptual level. This way, it was possible to test the effect of self-selected interval judgments on both a perceptual illusion (SWI) and accuracy.

## Self-selected intervals

The rationale for using the method of self-selected intervals is that people are not necessarily certain about quantities they may need to estimate, and may over time provide varying judgments of the same object due to uncertainty and noise in their perceptions. By letting participants select the boundaries of uncertainty freely, a more accurate measure of their true experience may be obtained [3].

To clarify the terminology in the context of this study, people estimate the weight of an object, and by reporting that estimate they have made a judgment. In other words, an estimate is what a person perceives and/or thinks, and a judgment is what that person explicitly reports as their estimate. The self-selected intervals (SSI) method has previously been investigated in studies asking for willingness to pay (WTP) [3] and the statistical properties of the method have been investigated as well [2, 6]. However, SSI has not previously been investigated with a task that has an objectively measurable and correct answer. Because weight is objectively measurable in grams, while WTP is subjectively determined by each individual, additional insights can be gained. Furthermore, we have added the size-weight illusion to the design, which adds a perceptual layer to the task. This enables us to further understand the usefulness of the self-selected intervals.

The self-selected interval judgments process was designed as follows. A participant estimated the weight of a box in grams, initially providing a numerical interval that the participant felt that it included all reasonable values of true weight. In the next step, this interval, for example, 200 to 400 grams was split in two equally wide intervals, 200–300 and 300–400 in this example. The participant was asked to choose which of the new smaller intervals was more likely to include the true value. The procedure of splitting the interval in half was repeated three times to successively converge towards the participant's best estimate of the box's true weight. Hence, the final interval midpoint should be a participant's best estimate, but this point was never judged explicitly by the participants.

Other previous research, using other types of interval judgments, has aimed at indicating how confident a participant was in providing the correct answer. With this type of intervals, if

a participant states intervals with 90% confidence, 90% of the intervals should include the true value [7]. Teigen and Jørgensen asked participants to give intervals which should include the correct answer of general knowledge items with some degree of confidence, e.g., 90%. In general, such intervals are found to be too narrow, illustrating that people are more confident about their knowledge than motivated. When [8] replicated this result they also found that participants produced intervals that were very similar even though the levels of confidence were different. This led to severe overconfidence for 90% intervals and less overconfidence for 50% intervals. When participants were asked to produce intervals without a specified confidence level overconfidence was reduced, but not eliminated. Furthermore, participants judged all outcomes that fell within an interval given by others (experts) as equally likely to be correct. This shows that interval judgments are not easy for people to calibrate in respect to their knowledge, but they are a powerful tool for understanding human judgment. In this study, our interval method used the midpoints of successively converging intervals to determine a participant's best estimate. This mitigates the effect of overconfidence leading to narrow intervals. To exemplify, the judgments 400g and 200g have the midpoint 300g and even if a more confident person would instead answer with 350g to 250g, the midpoint would still be 300g.

## The size-weight illusion

Augustine Charpentier [9] was the first to investigate the SWI experimentally. As an example of how strong the illusion can be, Charpentier showed that for two copper balls weighing 266 grams each, 200 grams had to be added to the larger ball to make it feel that it had the same weight as the smaller ball [9]. The illusion persists even when objects are held without being seen [10] or pushed instead of held or lifted [11], but the strength of the illusion can be moderated by the style of lifting [12]. The SWI has been thoroughly investigated in many ways, for example by measuring expectations of size or volume without touching the objects [13], and has been suggested to arise due to competing prior beliefs about relative volume and density between objects [14].

Most SWI-studies use some subjective measure of heaviness (e.g. magnitude estimation), see Saccone, Landry, & Chouinard (2019) for the first meta-analysis on the topic [4]. In our investigation of self-selected intervals, we asked the participants about the true weight of different boxes, rather than their subjective perception of weight. By asking for the weight in grams, we could test if interval judgment midpoints are more accurate than point estimates. In addition, we can find if the intervals will include the true values. We were also able to test if the SWI holds for estimates of true weight.

SWI has previously been found to hold for judgments of true weight in an exercise study [15], and we will try to replicate that finding as well. Furthermore, the SWI is a reliable illusion to include in an experiment of method comparisons because it is resistant to factors such as whether or not the weights are lifted simultaneously [16], and adjustments of the sensorimotor system due to repeated lifting [17, 18]. Furthermore, focusing on weight, rather than size, has been found to increase the illusion [19]. By asking for the true weight of objects participants should have this as their primary focus and give a strong illusion. This should facilitate finding potential differences between the methods.

## Summary of study aims

The overarching purpose of the present study was to investigate how well self-selected intervals can be applied to a weight judgment task, and compare it to the traditional method of point judgments. First, we wanted to verify that the size-weight illusion applies to judgments of physical weight in grams in the same way as it applies to pure weight perception measured with

magnitude estimation methods. Then, we wanted to compare the accuracy of estimates generated by the interval and point judgments; which method generates estimates closest to the true weight of an object? We also wanted to compare the strength of the illusion across the two different judgment methods. Finally, we wanted to find out if the self-selected interval method can help participants to converge towards their true best estimate. We wanted to know if the intervals, and the consecutive splits, result in more consistent estimates within each individual than do point judgments. In general, we showed in the experiment of this study that the interval method may have some benefits for measurements of human judgments of weight, and can reduce the size-weight illusion in some cases, but that there are limitations to the potential benefits of the method.

## Materials and methods

### Participants

A total of 32 participants were recruited from Studentkaninen.se (a participant pool for experiment participation at Stockholm University) and among students at the Department of Psychology, Stockholm University. The average age was 31 years (SD = 9). There were 15 men and 17 women, of whom 28 were right-handed and 3 left-handed; one participant did not report a dominant hand. The experiment involved two experimental conditions, one for each judgment task, point judgments and self-selected interval judgments. The two conditions were separated by at least two days. One participant did not show up for the second condition and was excluded from the analysis. Participants were rewarded SEK 200 (about € 20) in vouchers that could be used in a wide variety of stores. Furthermore, there was an extra reward of SEK 500 (about € 50) in vouchers to the participant who gave estimates closest to the correct weights, and the second closest participant was rewarded an extra SEK 200.

### Design

The experiment had a factorial design with the within-subject factors size of box (3 levels) and weight of box (5 levels per box). The weights were judged with point estimates and self-selected intervals on separate days. A detailed description follows in the sections: stimuli, judgment methods, and procedure.

### Stimuli

The stimuli consisted of 15 cardboard boxes of 3 different sizes, thus 5 boxes of each size. The boxes were (approximate) cubes with side length (cm) of 6.8 (small box), 12.3 (medium box) and 22.4 cm (large box). Counting the lid of the boxes the heights were 7.4 (small box), 12.8 (medium box) and 22.8 (large box). The size of the boxes made it impossible to make the heaviest small box as heavy as the heaviest large box. For this reason, only weight ranks 3–5 were represented in all three sizes. To center the entire stimulus set around a parameter the density 0.33 was represented in all three sizes, although density was not a main research interest for this study. Table 1 gives weight and density of each box, see also photo in Fig 1. Densities are calculated by using the volume including the height from the lid (e.g. 6.8*6.8*7.4 for the small box).

To increase the weight of the boxes, balance weights were glued alongside the inside walls in a symmetrical pattern. By spreading out the weights along the walls in a symmetrical pattern the boxes felt as a uniform weight, even when tilted slightly. The balance weights were 5g and 10g cuboid pieces of iron, and strung together by adhesive tape. To increase the stability of the large boxes, wooden sticks were placed as a cross, horizontally, in mid height of the box. The

**Table 1. Weight and density of the boxes used as stimuli in the experiment.**

| | Weight [g] | | | Density [g/cm$^3$] | | |
|---|---|---|---|---|---|---|
| Weight rank | Small box | Medium box | Large box | Small box | Medium box | Large box |
| 1 | 114 | | | **0.33** | | |
| 2 | 235 | 235 | | 0.68 | 0.12 | |
| 3 | 455 | 455 | 455 | 1.31 | 0.23 | 0.04 |
| 4 | 649 | 649 | 649 | 1.87 | **0.33** | 0.06 |
| 5 | 1140 | 1140 | 1140 | 3.29 | 0.58 | 0.10 |
| 6 | | 2199 | 2199 | | 1.12 | 0.19 |
| 7 | | | 3808 | | | **0.33** |

The density was the same for three of the boxes, the lightest small box, the medium heavy medium box and the heaviest large box, illustrated by bold density numbers.

large boxes were further reinforced with duct tape on the inside of the corners. To adjust box weights with the precision of a tenth of a gram the boxes were filled with cotton. In cases where the final 5g and 10g weights could not be symmetrically fit on the walls, single weights were embedded in the center of the cotton (e.g. if 10g was needed it could only be split up on 2 x 5g weights that could not be split symmetrically around 4 walls). See Fig 2 for an illustration of the boxes' construction.

## Judgment methods

The weights of the boxes were estimated by the participants and reported by *point judgments* and *self-selected interval judgments*. Each judgment method was conducted in a separate session. The instruction for *point judgment* was simply to judge the true weight of the boxes in grams or kilograms.

The instruction for *self-selected interval judgment* was to first give an interval (two numbers) representing the smallest and the largest weight the box reasonably could have (i.e. it would be

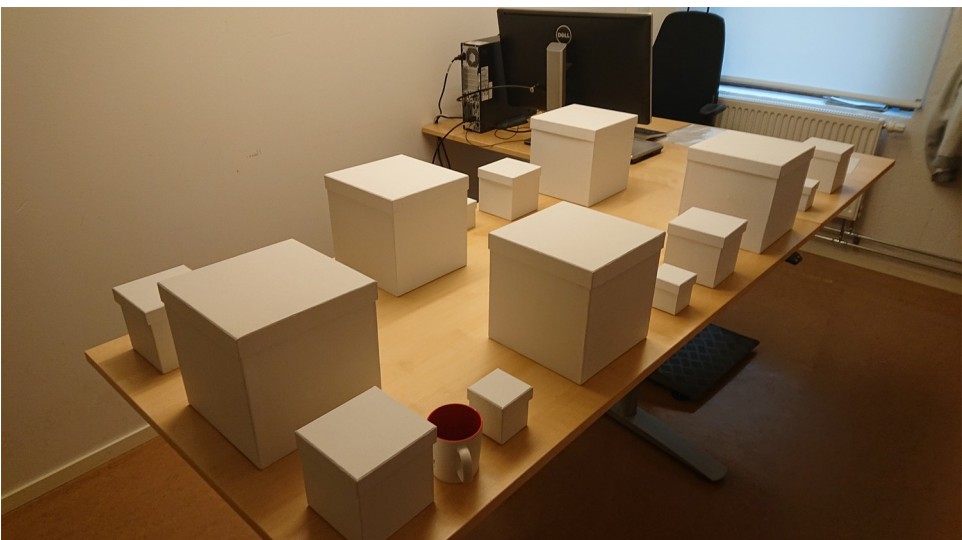

**Fig 1. Photograph of the arrangement of the 15 boxes before the first session.** Participants were asked to judge the weight of the mug, shown in the photo, before the experiment as an introduction to the lifting task. The experimenter sat behind the computer screen at the back and recorded the responses from the participant as they estimated the weight of the boxes one by one, starting with the medium box at the top right of the photo.

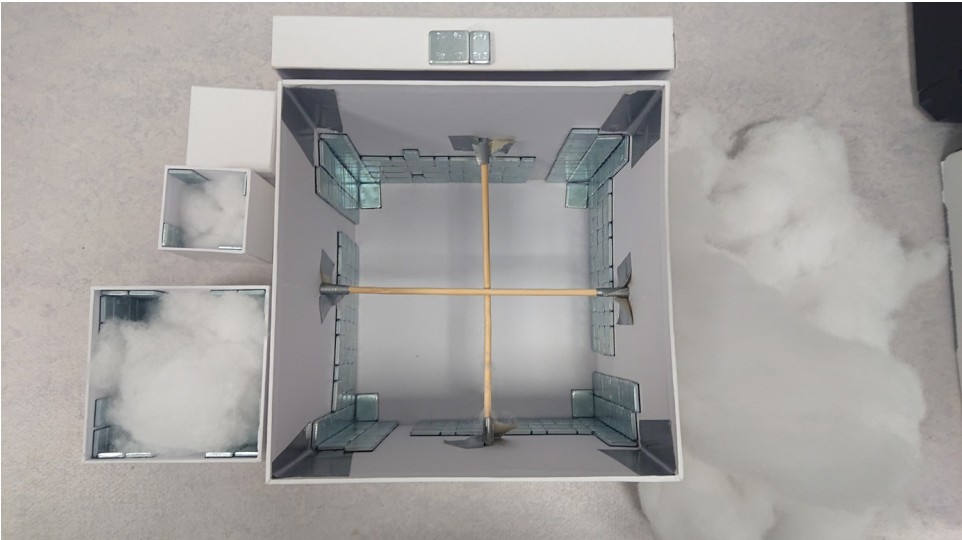

**Fig 2. Photograph of the inside of the boxes and the added weights.** The photograph illustrates examples of how the box weights were adjusted in each of the three different sizes used in our experiment. Single 10g and 5g weights can be seen on top of the lid at the top of the photo.

unreasonable to the participant that the true weight was outside the limits of the interval), and then answer three follow-up questions. The first follow up question asked if the box weighed less or more than the arithmetic midpoint of the first interval. The binary answer to this question was used to define a second interval half the width of the first interval, and this procedure was repeated two more times to generate a third and a fourth interval. The midpoint of the fourth and final interval was used as the estimate of a participant's best guess. This example illustrates the procedure:

1. A participant judged the weight of a given box to be between 100 and 200 g. Thus, the first interval was [100, 200].

2. The participant was then asked whether the box was lighter or heavier than 150 grams (midpoint of first interval), and answered 'heavier'. From this answer, a second interval was derived: [150, 200].

3. The participant was then asked whether the box was lighter or heavier than 175 grams (midpoint of second interval), and answered 'lighter'. From this, a third interval was derived: [150, 175].

4. Finally, the participant was asked whether the box was lighter or heavier than 162 grams (midpoint of third interval, rounded to closest even integer), and answered 'heavier'. From this, a fourth interval was derived: [162, 175].

## Procedure

Prior to the experiment, the participants signed an informed consent to participate in the study. As a general introduction to the experiment, the participant was asked to pick up a mug (249g) from the table and assess its weight. Furthermore, this gave us an indication of whether a point or interval judgment would be most natural for weight to report estimates (it turned out that the large majority gave a point judgment).

The participants were then given a written instruction describing the box weight estimation procedure. There was one instruction for the point judgment task and another instruction for the self-selected interval judgment task.

For the lifting part of the procedure, the participants were instructed to first lift the boxes with two hands, and then place the box in the palm of one hand. The experiment leader illustrated a motion where the box was lifted and held with the elbows at approximately a 90-degree angle, for both the two hands and one hand grip. After holding the box with two and then one hand, as instructed, the participants were allowed to hold the box in any way (and how long) they liked. The only restrictions for handling the box was that it was not allowed to shake, or tilt, the box in a way that could compromise the integrity of the box, or open the box. While holding the box they gave their estimate of the box's weight as either a point or an interval.

The 15 boxes were randomized into three different sequences (orders of the boxes). There were three sessions, one sequence per session. The first sequence of 15 boxes was set up before the participant entered the experiment room for the first time. After judging each of the 15 boxes once, in the order of the presented sequence, the participant was asked to leave the room while the experimenter set up the second sequence of the same 15 boxes. However, the participants were not informed whether or not it would be the same set of boxes for the second session. The participant then entered the room and for the second sequence. This was repeated for the third and final session. The three session sequences were the same for the point and interval conditions.

In the point estimate condition, the experimenter entered the judgment into the computer and asked the participant to move on to the next box. In the interval condition, the experimenter entered the interval judgments. After entering the interval judgments, the computer returned the midpoint of the interval (arithmetic mean rounded to the closest even integer) and the experimenter asked the participant if the box was lighter or heavier than this midpoint. The experimenter entered the answer into the computer, which then, again, returned the midpoint of the new, halved, interval and the procedure was repeated.

The participants participated in the point and interval judgment conditions on two different days, with at least two days in-between. The order of the conditions was randomized among the participants, half of the participants started with point judgments and the other half with interval judgments.

After having completed the box-lifting task during the second day, the participants filled out a questionnaire regarding reference weights. They were asked if they, during the weight estimation task, thought of any weight(s) they were familiar with from before the experiment, and if they used some specific strategy when they made their weight estimations. Finally, the participants were asked to judge the volume in cubic centimeters ($cm^3$) of the three different box sizes.

## Statistical analysis

The main tool for statistical analysis was R v.4.0.3 [20] with RStudio v.1.3.1093 [21] including the packages nlme v.3.1–149 [22], lme4. v.1.1–26 [23] and *sjPlotv*.2.8.7 [24], Jamovi 1.6.23.0 [25] was used as a secondary tool. The data files and scripts for the analyses can be found at https://figshare.com/s/92fc9e629e2a3b2a2343

## Results

The result section is divided into three main parts, the size-weight illusion, accuracy of weight judgments, and stability of judgments. Within each part we will first examine the point

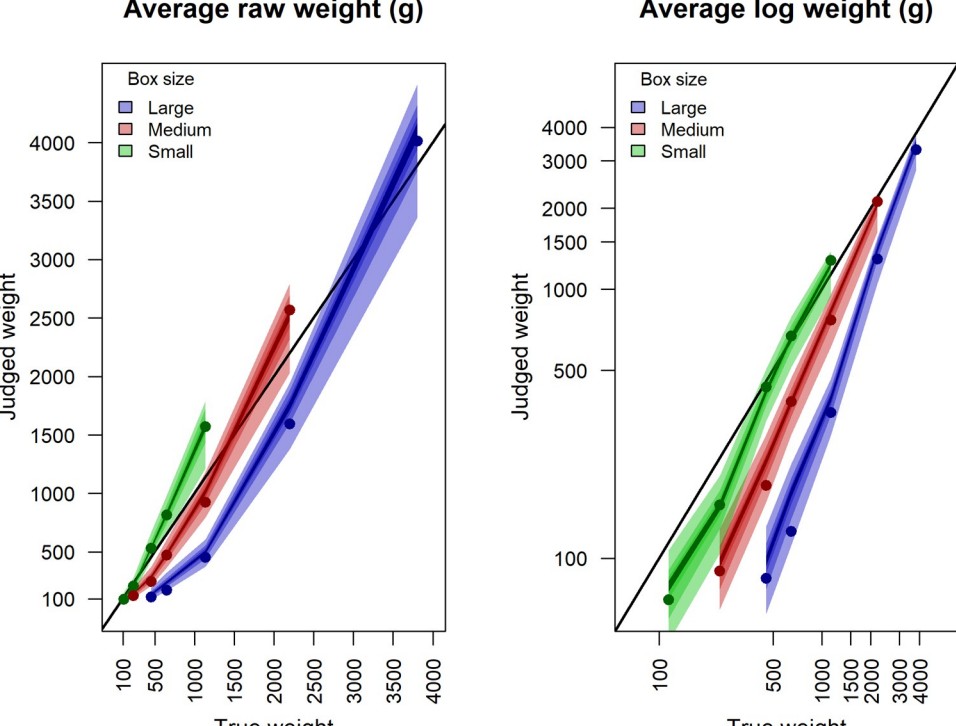

**Fig 3. Average judgment of each box's weight.** Points illustrate average point estimates and the colored fields boundaries are determined by the average upper boundary judgments and lower boundary judgments of the self-selected intervals. The colored fields become darker with each consecutive split of the intervals. Hence, the lightest color indicates the average width of the first interval (widest) and the darkest color indicates the average width of the fourth and final interval (most narrow). The aggregate results are stable across sessions, shown in the appendix with plots similar to Fig 3, but for each separate session. Importantly, there was great individual variability in the judgments, shown with figures for each participant, in the appendix.

judgments, then the self-selected interval judgments, and finally compare the two measures. The main size-weight illusion issue is, to what extent that boxes that weigh the same but are of different size are judged to weigh differently. The main accuracy of judgment issue regards how close judgments are to the true physical weight. The main stability of judgment issue regards how consistently a person judges the same box across different sessions.

It is important to note that complete accuracy means that there is no illusion, because in that case, boxes of the same weight would be judged to weigh the same independent of box size. However, a reduced size-weight illusion does not automatically mean better accuracy. For example, if the judgments of a small and large box were 500g and 600g the average would be 550g. The same average would be obtained by the judgments 400g and 700g. If the true weight was, for example, 300g, the first and second pair of judgments would deviate 250g on average from the correct weight, but the second pair of judgments (400g and 700g) would mean a greater size-weight illusion than the first pair (500g and 600g).

To summarize our data Fig 3 illustrates the aggregate response pattern. The aggregate results are presented here primarily as a descriptive tool to give a general sense of this experiment's results as a whole; detailed analyses will follow section by section. The diagonal line is a reference line for correct responses. The distance between box judgments of the same weight illustrates the size-weight illusion (SWI). Note that the colored fields illustrate the average range of the interval judgments and the points the average point judgments. For each

consecutive split the colored fields become darker, thus, when a point falls on the darkest area of the field it indicates that it falls within the limits of the fourth and final interval, which was 1/8 of the initially self-selected interval. Fig 3 also illustrates that the responses increase exponentially as a function of weight for all box sizes, and that the average judgments show a linear trend when they are log transformed. For this reason, the log transformed values were used in our linear models comparing accuracy.

### The size-weight illusion

**Point estimate.**   First, we wanted to confirm that our experiment replicated the size-weight illusion (SWI). We calculated the average point judgment for each of the boxes. Averages were calculated across all participants and trials, and Fig 3 shows a clear general size-weight illusion. Hence, we have shown that the illusion holds for judgments of actual weight of an object. Judgments of actual weight is a somewhat different task to the task in most studies of SWI, which commonly use magnitude estimation, and sometimes reference weights [4]. The distinction is important because judging the true weight directly does not explicitly lead the participant to make direct comparisons of specific boxes' weights. However, it should be noted that written subjective reports after the experiment showed that most participants thought of one, or several, reference weight(s) while making their judgments; for example, a carton of milk of 1 liter which weighs approximately 1kg.

To further check for the robustness of the size-weight illusion we counted the number of cases in which, within a session, a smaller box was judged as heavier than the box one size larger of the same true weight (i.e. for each weight, small was compared to medium and medium compared to large). We will refer to these as SWI-cases. The comparisons were made only within each session (in each condition there were 3 *sessions* with 15 trials). To clarify, the weights 235g, 455g, 649g and 1140g were present both as small and medium boxes, generating 4 comparisons of boxes of the same weight, but different size, per session. The weights 455g, 649g, 1140g and 2199g were present in both medium and large boxes, generating another 4 comparisons per session. This means that the theoretical maximum number of SWI-cases per participant was 24, for each type of judgment. The lightest weight (114g) was only present for the small box, and the heaviest weight (3808g) was only for the large box, thus these weights could not be compared across sizes.

The number of observed SWI-cases per participant ranged from 16 to 24 with a median of 23, and the mode was 23 as well (11 of 31 participants). Only 7 participants showed fewer than 21 SWI-cases (Q1). Note that 12 SWI-cases are expected by chance if a participant judges the weights at random. This applies also if a participant experiences no illusion, because of lack of consistency (identical judgments occurred only in 31/744 = 4.2% of cases). Summing all SWI-cases across participants showed that boxes were judged heavier than the one size larger counterpart 90% of the time (667 of 744 comparisons).

**Self-selected intervals.**   The robustness of the SWI holds for self-selected intervals and each of the consecutive splits, which is illustrated by the group data in Fig 3. The figure illustrates that average intervals do not overlap between boxes of the same weight, which illustrates the strength of the SWI, on the aggregate level. Non-overlapping intervals were found also within individuals. If a participant perceived that two boxes could not possibly weigh the same, the 1st intervals (the widest) of those two boxes should not overlap. We calculated the number of cases the lower interval limit of a smaller box was judged larger than the upper interval limit of the one size larger box of the same weight. Interval limit comparisons were carried out within each participant and session, as for the SWI-case analysis with point judgments. The average number of cases that a participant found it unreasonable that the boxes

**Table 2. Descriptive statistics of SWI-cases per participant for the interval conditions first and final mid, as well as the point judgments.**

|  | Min | Q1 | Median | Q3 | Max | Mode | Mean | SD | Sum | % |
|---|---|---|---|---|---|---|---|---|---|---|
| Point | 16 | 21 | **23** | 23 | 24 | 23(11) | 21.52 | 2.41 | 667 | 89.65 |
| Mid1 | 13 | 19.5 | **21** | 22 | 24 | 22(9) | 20.32 | 2.97 | 630 | 84.68 |
| Mid4 | 14 | 19 | **22** | 23 | 24 | 22(7) | 20.71 | 2.84 | 642 | 86.29 |

Number of participants per mode in parentheses (out of n = 31). Percentage of total SWI-cases was calculated as the sum of SWI-cases divided by 744 (the total number of comparisons).

could weigh the same was 12.29/24 (51%), the range was 2/24 to 23/24, the median was 11/24 (46%), and the mode (5 participants) was 19/24 (79%). Note that participants choose the limits of their intervals freely (50% would have been the prediction for random responses to the question "is it possible that the boxes weigh the same, yes or no?").

**Comparisons of point judgments and interval judgments.** Even though the interval measures illustrate the strength of the SWI, we wanted to know if the influence of size on judgments of weight could be reduced with the self-selected interval method. To make interval judgments comparable to point judgments we used the midpoint of the intervals as a metric. Because the interval splitting procedure was successively converging, we will focus the following analyses on the first interval midpoint, called mid1, and the fourth and final interval midpoint, called mid4.

To calculate the number of SWI-cases for interval midpoints we used the same procedure as with the point judgments. That is, we calculated the number of times each participant's estimate of a smaller box was greater than a box one size larger of the same true weight, descriptive statistics in Table 2.

First, we compared the number of SWI-cases per participant between point judgments and interval mid-points. We found that of the 24 comparisons per participant, the average number SWI- cases was for mid1 20.3 ($SD$ = 3.0), and for point judgments 21.5 ($SD$ = 2.8). Hence, the number of SWI-cases per participant was on average 1.2 less for interval mid1, compared to point judgments, $t$ = -2.7, $df$ = 30, $p$ = 0.012 (two-tailed), 95% $CI$ [-2.1, -0.3].

To further analyze the effects of size on weight judgments depending on judgment type, we compared the strength of the SWI between point judgments and interval mid1 and mid4. The metric for SWI strength was calculated as the difference between weight judgments of a box and its one size larger counterpart which was divided by the lighter of the two. This measure of SWI will be referred to as the SWI-factor from here. To exemplify, if a small box was judged to weigh 500g and the medium box 400g, the SWI-factor is (500–400)/400 = 0.25 (25% heavier smaller box). If instead the medium box was judged the heavier, in the example above, we would get (400–500)/400 = -0.25 (25% lighter smaller box). There were 8 comparisons per session which gives a total of 24 comparisons per judgment method and participant. We found some large outliers, especially in the point condition where one SWI-factor was as large as 49. We excluded SWI-factors greater than 10 and smaller than -10. The number of outliers removed was 16 for the point measure, 12 for the mid1 and 13 for mid4, that is, approximately 2% of the 744 SWI-factors originally computed per condition were excluded. See Table 3 for descriptive statistics before and after removing the outliers.

Because, the weights were different for SWI-factors of small/medium comparisons (weights 235g – 1140g) and medium/large comparisons (weights 455g – 2199g), We analyzed the data separately for each of these factors. We also excluded outliers and used only the data described in the bottom half of Table 3. A repeated measures ANOVA with the factors 2 (measure: point or mid1) x 4 (weight) x 3 (trial session) showed a significant main effect of measure for both

**Table 3. Descriptive statistics of weight judgments of a smaller box proportional deviation from the corresponding box that was one size larger.**

| Measure | Min | Q1 | Median | Q3 | Max | Mean | SD |
|---|---|---|---|---|---|---|---|
| Point | -14.00 | 0.33 | 0.86 | 1.69 | 49.00 | 1.69 | 3.70 |
| Mid1 | -36.00 | 0.21 | 0.67 | 1.60 | 25.67 | 1.24 | 2.95 |
| Mid4 | -39.91 | 0.23 | 0.71 | 1.58 | 24.50 | 1.25 | 3.06 |
| *Filtered, outliers -10 > SWI-factor > 10 removed* | | | | | | | |
| Measure | Min | Q1 | Median | Q3 | Max | Mean | SD |
| Point | -1.09 | 0.33 | 0.87 | 1.62 | 9.00 | 1.33 | 1.66 |
| Mid1 | -4.00 | 0.21 | 0.67 | 1.50 | 9.00 | 1.09 | 1.49 |
| Mid4 | -4.03 | 0.22 | 0.71 | 1.51 | 9.68 | 1.08 | 1.43 |

Positive values indicate how much heavier, proportionally, a smaller box in a same weighed pair of boxes is judged (i.e. 0.86 means 86% heavier). Negative numbers indicate the same but the larger box was the one judged as heavier (reverse-SWI).

the small/medium comparisons $F(1,30) = 5.636$, p = 0.024, $\eta^2_p = 0.158$, and medium/large comparisons $F(1,30) = 6.275$, $p = 0.018$, $\eta^2_p = 0.173$. This indicates that SWI is lower with interval judgments compared to point judgments. However, the SWI-factor peaked for the point measure, while it remains at a similar level for lighter and heavier weights for the interval mid, illustrated by Fig 4. This peak was found at the second weight in rank order for both the small/medium comparisons (455g) and the medium/large comparisons (649g). The interaction effect between weight and measure was significant for the medium/large comparison $F(3,90) = 4.08$, $p = 0.009$, $\eta^2_p = 0.120$, but not quite for the small/medium comparison $F(3,90) = 2.49$,

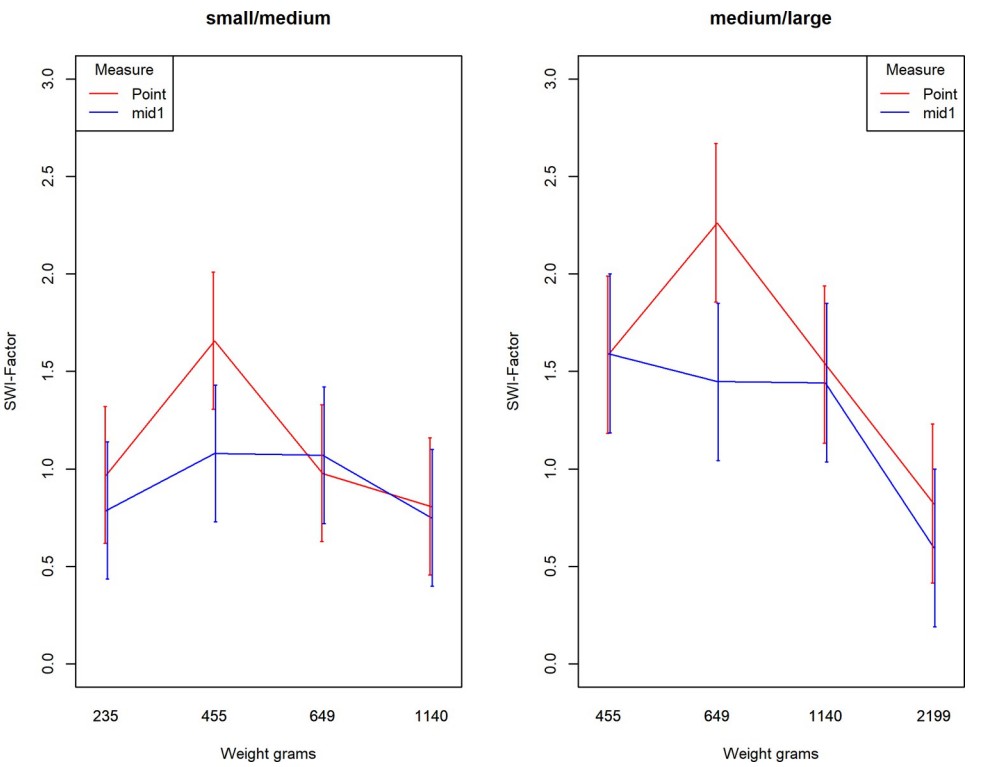

**Fig 4. Average SWI-factor depending on weight and judgment method.** Separate lines for SWI-factor for small/medium (solid) and medium/large (dotted) comparisons. Whiskers indicate 95%-CI's.

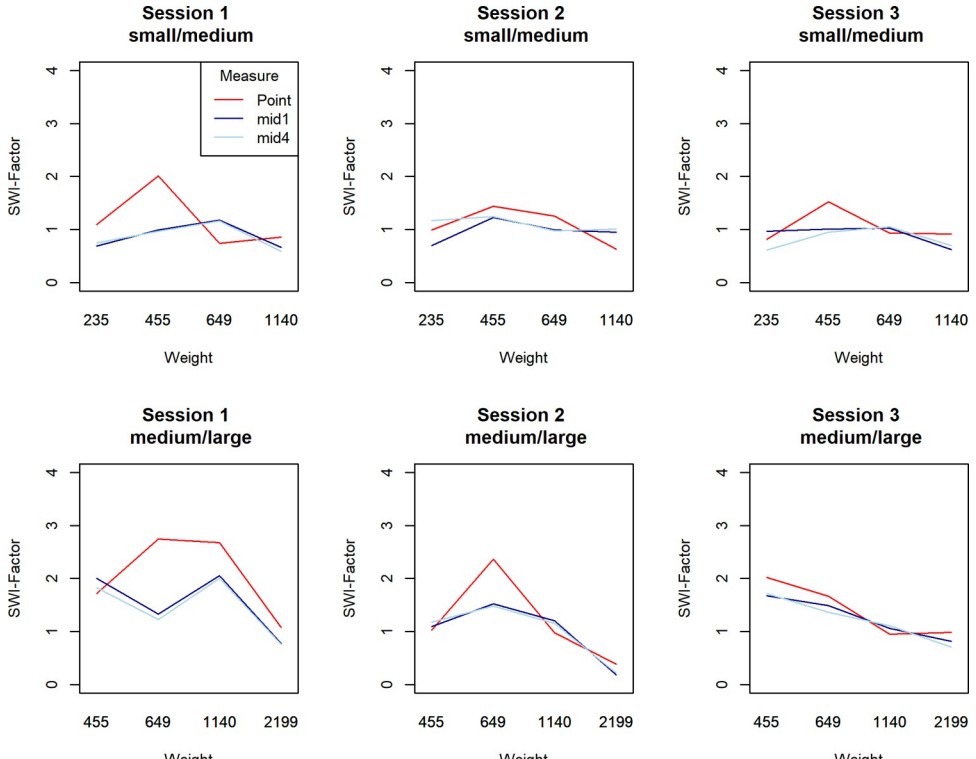

**Fig 5. Average SWI-factor depending on weight and judgment method for each judgment session.** Separate lines for SWI-factor for small/medium (solid) and medium/large (dotted) comparisons.

$p = 0.065$, $\eta^2_p = 0.077$. We had no a priori hypothesis about this specific peak so this finding should only be considered as exploratory. Importantly the main effect should be interpreted cautiously in the presence of a significant interaction [26].

Our interpretation is that the interval method potentially may reduce SWI, but only by limiting responses indicating large illusions rather than reducing the illusion in general. In other words, the point judgments peak that is absent in the interval condition may imply that the interval method puts an upper limit to the illusion. The pattern of a peak at the second lightest weight holds across all three sessions, with the only exception that in session 3 the lightest medium/large comparison (455g) led to the greatest SWI-factor, implying some reliability of this finding. Fig 5 illustrates the average SWI-factor across sessions. Mid4 was included to show that there is no clear reduction (or increase) in SWI-factor following the splitting procedure (narrowing the intervals), and we did not analyze the SWI-factor for mid4 further for this reason.

To summarize, the size-weight illusion is very robust, and holds to the extent that judgment intervals often do not even overlap for boxes of the same weight but different size. The interval method seems to reduce illusions above a certain level, that is, the interval judgment method puts a limit on the measured illusion.

## Accuracy of weight judgments

To quantify the accuracy of judgments we calculated the proportional absolute deviation, *pad*:

$$\left| \frac{j}{w} - 1 \right| = pad \tag{1}$$

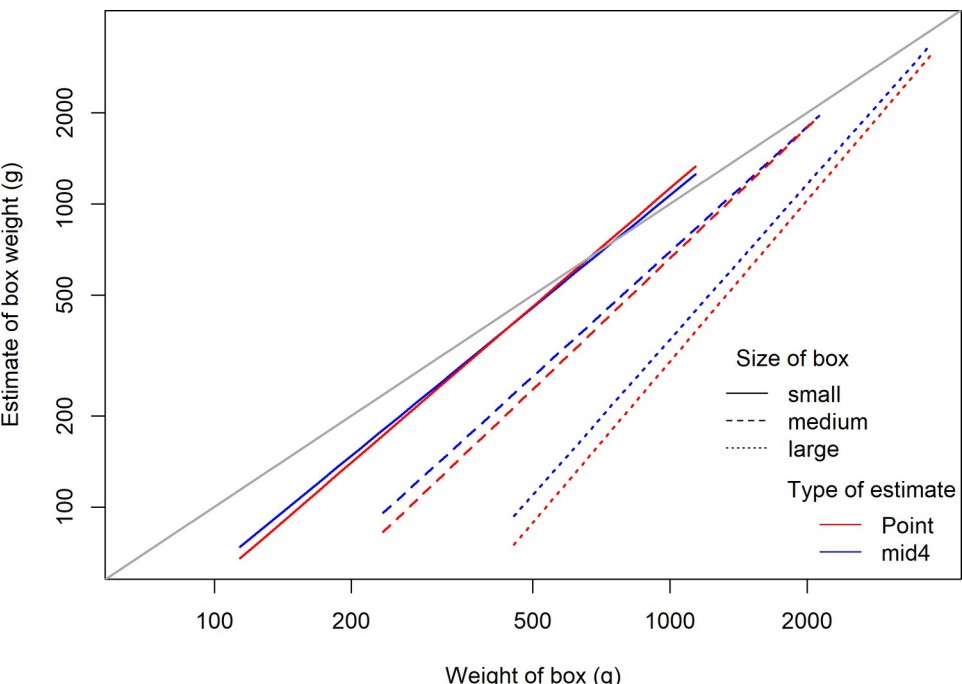

**Fig 6. Slope estimates for log of point judgments and log of mid4 in grams.** Main effects: the log of the true box weight, the type of judgment estimate (point vs mid4), and size (small, medium, large). The model also contains first-order interaction effects, and random effects of, log of true weight, type of judgment estimate, and size. The numerical model estimates can be found in S1 Table.

where *j* is a participant's estimate (e.g. point judgment or one of the midpoints of an interval) and *w* is the true weight of the judged box. To exemplify, for a box of weight *w* = 200, judgments of *j* = 100 and *j* = 300, would both result in a *pad* = 0.5. The average pad was for point judgments 56.5%. For interval judgments, the average pad was for the first interval midpoint 56.0% and the fourth and final mid 55.6%. The average pad became marginally but monotonously smaller for each split, but the 0.4 percentage point reduction from first to final mid was not statistically significant, and neither were point judgments, in dependent two-tailed t-tests.

To estimate and compare the accuracy of point judgments to the accuracy of the final midpoint of the intervals (mid4), across the range weights and sizes of boxes, the logarithm of the measures was modeled by a mixed effect model using restricted maximum likelihood estimation. The logarithm was used because then the response pattern for each box becomes approximately linear. This is illustrated by Fig 3. We followed the modeling strategy suggested by Zuur et al. [27] to give less biased estimators of the variance components. Therefore, our final model uses restricted maximum likelihood estimation.

The results indicate that the final interval judgments (mid4) are marginally closer to the correct judgments compared to point judgments. The model results are illustrated in Fig 6 (numerical estimates can be found in S1 Table) which also shows that all fixed-effects were significant. The figure clearly shows the size-weight illusion in both conditions. The slopes become steeper with increasing box size indicating that the largest box was the hardest to judge accurately within the range of weights that we used in this experiment. Furthermore, the difference between the lines for point judgments and mid4 is greatest for the largest box, indicating that the interval method may improve accuracy more when the task becomes more difficult.

**Further exploratory analysis of the self-selected interval accuracy.** To further investigate the accuracy of the intervals we calculated how many intervals that included the true value, we call this a "hit". A hit was defined as the true weight being smaller than the upper limit and greater than the lower limit reported. Of the 1395 intervals (31 participants * 45 judgments) only 290 intervals hit the true value. On average participants hit the true value with the interval 9.35/45 times, a hit rate of 20.8%, $SD = 19.3\%$. The least accurate participant had 0% hits and the most accurate had 75.6% hits. The hit percentage of a participant correlated strongly with participant average interval width, $r = 0.82$, 95% CI [0.65, 091], $t = 7.63$, $df = 29$, $p < .001$, and, interval width relative the true weight of a box, $r = 0.76$, 95% CI [0.56, 0.88], $t = 6.37$, $df = 29$, $p < .001$. Furthermore, hit rates were on average higher for small boxes (28.4%) compared to medium boxes (20.6%) and large boxes (13.3%).

It should be noted that the hit-rate for the intervals was approximately halved for each consecutive split of the interval, 1st interval 20.8%, 2nd interval 10.1%, 3rd interval 4.4%, 4th interval 2.3%. For each split, half the hit rate for half the interval width is expected by chance. In other words, when participants managed to select intervals that included the true value, they did not manage to keep the true value within their selected interval boundaries any more than if the split intervals were selected at random.

We also calculated how often participants captured their own point judgment within the interval judgment of that same box in the corresponding session (judged upper limit $\geq$ point judgment $\geq$ judged lower limit). That is, a point within the limits of the corresponding interval judgment was coded as 1 (captured by the interval) and outside the limit as 0 (not captured by the interval). As mentioned, the sequence of boxes in a session (1 to 3) was the same in the point and interval conditions. Participants' average proportion of point judgment captured by the corresponding intervals was 42.5% ($SD = 19.2\%$). This is an important finding because it suggests that more often than not participants did not find their point judgments as reasonable weights according to the interval limits. The order of the tasks did not seem to affect how often point judgments were captured by intervals, 41.3% ($SD = 18.9\%$) for participants doing the interval judgments the first day and 43.6% ($SD = 20.1\%$) for point judgments the first day. This indicates that there were no calibrating effects of one type of judgment over the other.

To find out to what extent the interval width could explain the case when intervals covered the points; we first regressed points covered on interval width, across all participants' judgments, and found that only 1% of the variance in points covered was accounted for by the interval width.

To account for intervals being wider for greater weight judgments we regressed points covered on interval width relative to its midpoint (interval width/mid1, the relative interval width). The relative interval width accounted for 14% of the points covered. This indicates that intervals are not well calibrated towards the points judgments and may employ a different process.

We also wanted to know to what extent the average interval width could explain the proportion of points covered. Therefore, we also calculated the average interval width for each participant, $M = 347$g, $SD = 337$g, and average relative interval width, $M = 0.49$, $SD = 0.30$. Then, we regressed the proportion of points covered on the average interval width, $R^2 = 0.22$, $b = 2.65$, $F(1, 29) = 7.941$, $p = .009$ and average relative interval width, $R^2 = 0.64$, $b = 0.52$, $F(1, 29) = 50.73$, $p < .001$. This may indicate that some people adapted the range of their intervals to the magnitude of their point judgment and were more calibrated towards their point judgments. This is corroborated by the individual plots found in the appendix which shows that some participants increased the width of their intervals as boxes got heavier and judgments larger, while other participants used narrow intervals across the range of weights. Furthermore, the intervals and points may differ in regard to what judgment is reasonable to derive from our perceptions.

## Variation of judgments depending on condition

We wanted to know if individuals' weight judgments would vary less when they use intervals rather than points (independently of if the judgments were accurate or not). In other words, we wanted to find out if the judgments would become more consistent with the self-selected interval method. To be able to compare box judgment variation across measures, while taking into account individual variation of judged box weights across boxes, we calculated the coefficient of variation for each participant. We calculated the standard deviation of each individual's estimate (point judgment or interval mid) of each of the boxes and divided that standard deviation by the same individual's average estimate of that box. Hence, the relative standard deviation of weight estimates was calculated from each participant's judgments of each of the boxes in each of the conditions (point and interval). As an example, if a participant during the three sessions judged a box as 350g, 500g, and 650g ($M = 500$, $SD = 150$,) the relative standard deviation for that participant and box would be 150/500 = 0.3. This is compared to the relative standard deviation of the interval midpoint of that same box, for example, 400g, 500g and 600g ($M = 500$, $SD = 100$) which would give 100/500 = 0.2 (a reduction of 0.1 in proportional variation).

The average relative standard deviation of point judgments ($M = .321$) was higher compared to both mid1 ($M = .287$, $t_{30} = 1.89$, $p = 0.068$), and mid4 ($M = .280$, $t_{30} = 2.25$, $p = 0.032$), however, the difference was not significant (two-tailed t-tests) compared to mid 1. Averages are illustrated in Fig 7.

It should be noted that the average relative standard deviation (SD/M) was greater for the lower limit judgments ($M = 0.32$) than the upper limit judgments ($M = 0.28$). Furthermore, in absolute terms the upper limit judgments varied more for all sizes and weights. This indicates that variation was larger because the numbers were larger. This supports the use of a relative measure of variance for comparisons of variation across methods.

To summarize, the self-selected interval method slightly reduced the variation of weight estimates of the same object. The reduction of average relative SD from point (.32) to mid4

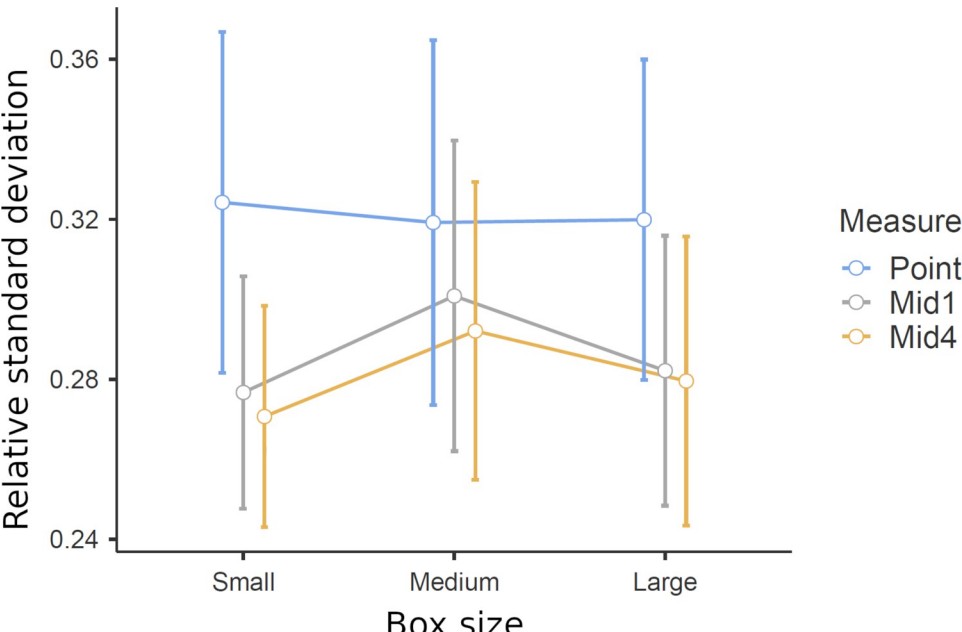

**Fig 7. Relative standard deviation of box judgments depending on measure and size of the box.** Whiskers indicate 95%-CI.

(.28) suggests a potential for the method. It needs to be developed further, specifically regarding the extent to which the process of successively converging intervals can be used to stabilize people's estimates of real-world parameters in an effective way.

## Discussion

In this study, we have investigated an interval judgment method called self-selected intervals [2, 3] and compared it to traditional point assessments, using a weight judgment experiment. The weights were boxes of different sizes, and the size-weight illusion was present during all trial sessions. Two potential strengths of the self-selected interval method were found. First, the self-selected intervals reduced some of the SWI (size-weight illusion), primarily when the point judgment SWI was at its maximum. Second, the midpoints following the successively converging intervals (splitting method) gave somewhat more stable weight estimates compare to point judgments; participants showed less intra-individual variation in their weight estimates derived from the judged interval (mid1), and following splitting procedure (mid4), than in their point judgments. However, the advantages of the interval method were small and we found only little support for the interval method as a way to get more accurate weight estimates. The differences between the estimates of the two methods were notably greater for both the largest cases of SWI and the largest box, which was the most difficult to judge accurately. This may indicate that the interval method becomes particularly useful when the task is difficult.

The interval method gave us a novel way to investigate the strength of the SWI. This is because interval judgments indicate the bounds on what a person would say is reasonable. Hence, no overlap between intervals indicates with greater certainty than point estimates that a participant differentiates between weights, even if the true weight is the same. For this reason, the interval method may be used as a new way to measure the strength of SWI. Furthermore, the interval methods reduction, compared to point estimates, of the largest SWI in the data set, may indicate that the measure can put a limit on the magnitude of the illusion. However, more research is needed to establish if the interval method actually reduced the illusion perceived, or if it is inherent in the method to limit extreme judgments in general, independent of what is judged.

Previous research has suggested that the illusion is impenetrable by cognition. This was shown by adding cognitive load to the task, and the illusion remained the same [28]. Assuming that interval judgments demand more cognitive effort than traditional point judgments, our results indicate that there may be a threshold for which illusions greater than that threshold may be limited by cognition. Furthermore, we do not yet know if it is the cognitive effort added to the task by using interval judgments or if judgments in general become less extreme with the method, as mentioned above. There is an important methodological difference highlighted here, because the study by Saccone et al. (2019) added cognitive load as a distraction while we added cognitive effort to the actual judgment process of the main task. If weight judgments are generally made based on sensory input, and little cognition is present to begin with, there is not much cognition to distract by adding cognitive load. On the other hand, if the experimental method elicits additional cognitive effort, this may start to moderate the resulting judgments or estimates given by the participants.

Earlier studies of the self-selected interval method have used measures of subjective preference, for example willingness to pay [3, 29]. In this study, judgments were regarded as objective true values, and judgments were most likely primarily based on perception, that is, information derived from sensory systems such as sight and haptics. Because we asked for the true weight of the boxes, a cognitive process was assumed to follow (and this assumption is

strengthened by the participants' retrospective reports which indicated that they thought about other objects' weight, such as a milk carton of 1kg, while trying to estimate the true weight of a box). All participants stated that they used a weight they are familiar with in real life as a reference for their judgments, with 1kg being the most common weight used as a reference, and a 1-liter milk carton being the most common object used as a reference object for that weight. The second most common objects were exercise weights. Importantly, some participants referred to only one weight, while others referred to several weights, or a range of weights. Reported reference weights ranged from 10g to 10kg. Whether or not the interval method is helpful or not for a person may depend on the range of references used, as this includes a wider base of prior knowledge used to determine one's judgment to respond with. For the present study we asked open ended responses for exploratory purposes. Because these responses were similar in nature, and generally clear, we encourage future studies to make the questions more specific so that these variables can be parameterized in an appropriate way to determine if some people may or may not benefit from judging with the self-selected intervals. Furthermore, the density of a participant's reference weights may have affected their weight judgments. S7 Fig shows that the density derived from the judgments deviates more from the true density the larger the box is. If the size affects the sensitivity to density, the reference weight may have different effects depending on its size compared to the judged box. To exemplify, a 1kg exercise weight disc may be perceived as more similar to the heavy small box while a 1kg milk-carton may be more similar to the medium box. Experimentally trying to manipulate what reference weights are used may give interesting information about accuracy of judgments and comparisons between the methods.

Another important factor to consider for both weight judgment and interval judgments is the amount of experience with lifting objects with known weights. A person with a lot of experience may use appropriate reference weights for the object lifted, and therefore be better at judging accurately. A person with a lot of experience may also be familiar with how difficult a task actually is, and therefore be able to calibrate their interval limits to properly reflect the uncertainty of their judgments. Hence, the experienced person may be able to narrow the judged interval down to a more accurate guess. Examples of people with experience in lifting weights, within the range used in our experiment, are people exercising with weights and people working in a post-office. It may be that the strong cognitive component in trying to estimate the true weight makes the average judgment accuracy similar between the two types of judgments (point and interval). The size-weight illusion, on the other hand, is primarily sensory driven, and the interval judgments did not reach the same level of illusion as the point judgments did.

Furthermore, the interval-method was found to increase the accuracy of judgment slightly, and this may be due to the judgments being based primarily on sensory information. For this reason, future research of the self-selected intervals can benefit from comparing cognitive judgments about the objective reality with perceptual experiences, as well as to preferences (such as WTP which can include a strong emotional component). The usefulness of the interval method may depend on the primary source of information, sensory, cognitive/theoretical or preferences based on cognition, emotion, or a blend of the two. Assuming that the interval-method can "zoom in" on a person's best, or true, response to a task, this process can be dependent on which mental systems are activated during that task.

## Limitations and future directions

While making interval judgments, participants generally held the box for a longer period of time than while making point judgments, because of the more elaborate way of responding.

Assuming that perceived heaviness increases with time due to increased muscle strain, this may have led participants to respond with somewhat heavier estimates in the interval condition. For further investigation of interval judgments of weight, the time a box is held should be standardized across conditions. Another aspect of this problem is that participants responded more times per box in the interval judgment condition, meaning that they may have adjusted simply because they responded several times. Hence, for future studies, to control for number of responses per item judged, there should also be a condition where participants respond several times for each box to account for the number of judgments made in the interval condition. Ideally, this can be compared to a condition with a single point estimate per box, but with the same amount of time holding the box (suggested above). This way the factor 'time holding an item' can be disentangled from the factor 'number of judgments' of that same item. This, in turn, would allow for an even more accurate comparison with the self-selected interval method. One problem with the repeated point judgment approach is that participants may perceive it as tedious, and/or unnecessary, to answer the exact same question several times consecutively. Therefore, the instructions should be very carefully worded, to avoid those problems. A suggestion may be to instruct participants to really think about the weight, and if their judgment of that weight is reasonable. This can also be used as a way to increase cognitive effort allocated to the task to see if there is a point where cognition can start to moderate the illusion (i.e. finding a threshold from which the illusion can be limited).

To gain more insight into the extent to which self-selected intervals are a useful tool in estimating and judging real world parameters, depending on if the task is perceptual, cognitive, or both, it would be useful to further investigate self-selected intervals with splitting in other contexts where the task is primarily cognitive, such as traditional judgment problems, and compare it to experiments primarily perceptive, such as magnitude estimation. We found that the successively converging intervals (splitting method), reduced the intra-individual variation of weight estimates to some extent, compared to point judgments. However, the estimates were stabilized within individuals only to a limited extent and a limiting factor may have been that participants understand "reasonable" in different ways (they were told to judge the greatest and smallest reasonable weight). Furthermore, we know from previous studies [7, 8] that people are generally too confident in interval accuracy. Many participants used very narrow intervals (see S1 Fig) which indicates that they had a very clear idea about what "reasonable" means and/or were overconfident in their ability to judge weights. Hence, working to standardize the procedure so that participants use the limits more similarly and ask for confidence judgments may help develop this particular method. We did not include confidence judgments because confidence was not a focal point per se. However, our data suggests that it may be useful in future developments.

Interestingly, even though the aggregate results (Fig 3) places point estimates close to the center of the intervals, when looking at each individual's judgments, point judgments fell outside the corresponding interval limits for the most part. This means that findings from studies where best point and interval judgments were made one after another within the same task session, e.g. [30], may not generalize to situations where only one way of assessment is used. Furthermore, having people specify where in the interval they think the best guess is with the splitting method (which is similar to defining a point in the interval), may not necessarily lead towards an estimate that conceptually corresponds to a point judgment. The point estimation process may be different than the cognitive process used for intervals.

It seems reasonable to assume that the natural reason for using intervals in real life is due to a specific estimation that is deemed as difficult to make. However, the objective difficulty of the task (accuracy) and the subjective difficulty of the task (confidence) are not the same, but these factors can provide insights into how the self-selected interval method compares to point

judgments. These factors may also be relevant for individual differences when intervals are used.

A way to gain a better understanding of how different people may use, and/or think in terms of intervals in real life, is to use cognitive process tracing methods. Such methods may find out what processes are involved in making point and interval judgments each one at a time or in conjunction with each other. This would probably be a very informative way to further investigate the usefulness of self-selected intervals.

To summarize, the self-selected interval method shows promise when studying a person's subjective estimates and numerical judgments, whether it is cognitive and preference driven as in previous WTP studies or perceptual and belief driven as in this study. However, there is still work to be done regarding standardization of the procedure and finding out how the interval interacts with judgments and estimates depending on the task's properties in the dimensions subjective/objective and cognitive/perceptual.

## Supporting information

**S1 Fig. Illustration (1 of 3) of each participant's average judgments for each of the boxes and judgment methods.** The figure is conceptually the same as Fig 2, which introduces the result section. Points indicate point judgments, intervals indicate the range of the upper and lower interval limits and the color darkens for each consecutive split (i.e. the darkest middle field indicates the upper and lower limits after the final interval split). True weights are described on the x-axis and judgments on the y-axis, thus, the diagonal line indicates true judgments. To these individual plots we have, in the title, added the number of size-weight illusion cases for Point (P), Mid1 (M1), Mid4 (M4) and no interval overlap (IO).
(TIFF)

**S2 Fig. Illustration (2 of 3) of each participant's average judgments for each of the boxes and judgment methods.**
(TIFF)

**S3 Fig. Illustration (3 of 3) of each participant's average judgments for each of the boxes and judgment methods.**
(TIFF)

**S4 Fig. Average judgments for each of the trial sessions.**
(TIFF)

**S5 Fig. Average judgments relative the true weight of the box for each of the sessions.**
(TIFF)

**S6 Fig. Average log judgments and log true weights.**
(TIFF)

**S7 Fig. Density derived from judgments relative to the true density of the boxes.** To illustrate how the judgments relate to density depending on box size we first computed the density derived from judgments by dividing weight judgments by the true volume of the judged box. We then calculated the derived judged density relative to the true density. This was done to be able to visualize the full range of judgments in a single plot. Furthermore, with this calculation, a value of 1.0 means that the derived judged density is the same as the true density. The left panel illustrates average judgments for point judgments (solid dots) and mid4 (empty square), along with the average judged upper and lower interval limits (vertical lines showing the range of the interval). The right panel illustrates linear regression lines fitted to the same data used

for the left pane. The solid lines are fitted to point judgments and the dotted lines are fitted to the mid4 judgments. The figure shows that the expected density for the different boxes are not the same because the regression lines intersect 1.0 on the y-axis at different points for each of the box sizes, large (blue), medium (red) and small (green). The average expected density for each box size is indicated by the value on the x-axis where the regression line goes through the horizontal line at 1.0 on the y-axis.
(TIFF)

**S1 Table. Model estimates: Mixed effect model illustrated in Fig 3.** True weight of boxes and judgment estimates were log transformed for this analysis. *Main effects* of true box weight, the type of judgment estimate (point vs mid4), and size (small, medium, large). *First-order interaction effects* between true weights, type of judgment estimate, and size. *Random effects* for true weight, type of judgement estimate, and size.
(DOCX)

## Acknowledgments

We would like to thank Angel Angelov for valuable input on the statistical procedures in this manuscript.

## Author Contributions

**Conceptualization:** Nichel Gonzalez, Ola Svenson, Magnus Ekström, Bengt Kriström, Mats E. Nilsson.

**Data curation:** Nichel Gonzalez.

**Formal analysis:** Nichel Gonzalez, Magnus Ekström, Mats E. Nilsson.

**Funding acquisition:** Bengt Kriström.

**Investigation:** Nichel Gonzalez.

**Methodology:** Nichel Gonzalez, Ola Svenson, Magnus Ekström, Bengt Kriström, Mats E. Nilsson.

**Project administration:** Mats E. Nilsson.

**Resources:** Mats E. Nilsson.

**Software:** Mats E. Nilsson.

**Supervision:** Mats E. Nilsson.

**Validation:** Nichel Gonzalez, Ola Svenson.

**Visualization:** Nichel Gonzalez, Magnus Ekström.

**Writing – original draft:** Nichel Gonzalez.

**Writing – review & editing:** Nichel Gonzalez, Ola Svenson, Magnus Ekström, Mats E. Nilsson.

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
