## [Decision Letter · Decision Letter 0]

22 Jul 2021

PONE-D-21-18240

Self-selected interval judgments compared to point judgments: A weight judgment experiment in the presence of the size-weight illusion.

PLOS ONE

Dear Dr. Gonzalez,

Thank you for submitting your manuscript to PLOS ONE. After careful consideration, we feel that it has merit but does not meet PLOS ONE’s publication criteria as it currently stands. Therefore, we invite you to submit a revised version of the manuscript that addresses the points raised during the review process.

While you will need to address all the concerns of the reviewers, the following concerns seem to be particularly important:

1) As the first reviewer points out, Buckingham et al. (2014) also had participants make point estimates of the weight of objects, so should be cited.

2) Please discuss to what extent density influences weight judgments.

3) As pointed out by the third reviewer, the literature review is incomplete. Please cite and discuss more work that examines the size-weight illusion. To address the concern of the second reviewer, this review should show that the issue you address is not a straw man and that numerical point estimates are common.

4) Please describe your participants in more detail – they may be unusually good at this task.

5) I agree with the third reviewer that there is a potential confound between the conditions – participants provided more estimates in one condition than in the other. This issue needs to be discussed.

6) Please indicate which approach should be used in which circumstances.

We look forward to receiving your revised manuscript.

Kind regards,

Piers D. L. Howe

Academic Editor

PLOS ONE

Journal Requirements:

3. You indicated that ethical approval was not necessary for your study. Could you please provide further details on why your study is exempt from the need for approval and confirmation from your institutional review board or research ethics committee (e.g., in the form of a letter or email correspondence) that ethics review was not necessary for this study? Please include a copy of the correspondence as an ""Other"" file.

5. Please ensure that you refer to Figure 6 in your text as, if accepted, production will need this reference to link the reader to the figure.

Reviewers' comments:

Reviewer's Responses to Questions

**Comments to the Author**

1. Is the manuscript technically sound, and do the data support the conclusions?

Reviewer #1: Yes

Reviewer #2: Yes

Reviewer #3: Partly

2. Has the statistical analysis been performed appropriately and rigorously? 

Reviewer #1: No

Reviewer #2: Yes

Reviewer #3: Yes

3. Have the authors made all data underlying the findings in their manuscript fully available?

Reviewer #1: Yes

Reviewer #2: Yes

Reviewer #3: Yes

4. Is the manuscript presented in an intelligible fashion and written in standard English?

Reviewer #1: Yes

Reviewer #2: Yes

Reviewer #3: Yes

5. Review Comments to the Author

Reviewer #1: General comments:

The study is not flashy but I appreciate that it reports a scientifically sound experiment and that these simple experiments that explore methodological issues are of value to the scientific community. I think the paper should be publishable but that some changes are required to increase clarity (see below re issues with the explanations of analyses and also figures) and also substance of the paper.

Specific issues:

Page 3, line 96. I believe this paper has assessed the SWI in terms of estimates of weight: Buckingham, G., Byrne, C. M., Paciocco, J., van Eimeren, L., & Goodale, M. A. (2014). Weightlifting exercise and the size–weight illusion. Attention, Perception, & Psychophysics, 76(2), 452-459.

I don’t think this invalidates the current study but it does reduce the novelty. Personally I think the study still has value but the authors need to adjust their statements throughout the manuscript with respect to this issue.

The meta-analysis that the authors cite (along with some other papers) strongly suggest that density influences weight judgments including the SWI. The authors report stimulus density but then do not report whether the manipulations impact the effect of density of weight judgments (overall accuracy as well as the SWI). I think this would be a useful addition to the paper and will make it more substantial and informative.

Minor issues:

Page 1, lines 34-40. “Another benefit in using this approach… in respect to both accuracy and illusion.” I do not understand what the authors are saying here. I think they need to unpack it more.

Page 2 lines 68-80. The relevance of this paragraph for the current study should be made clearer.

I think the authors need to explain why they included the average log estimates (Fig 2 right panel).

I find Figure 3 to be very unclear and think it needs to be revised. The lines and colours are difficult to see. Maybe use different colours for small/medium vs medium/large or else plot them separately. Also, I think it will be easier to understand if the x axis has the actual weights instead of weight rank (because weight rank is not at all intuitive). Likewise I think similar changes should be made to Figure 4 for the sake of readability.

Page 9 – I do not understand why the authors describe ANOVAs for the SWI-factor but then do not report them. This needs to be reported or at least explained more clearly.

Page 10, line 395 – what test was done to determine statistical nonsignficance? It should be reported.

Page 10, final paragraph – now I see that the log weights are analyses but again I do not know why. The authors need to explain this, as well as the maximum likelihood estimation analysis. Also, they refer to Figure 3 here – is this correct?

Page 11, line 424-5 – hit rates were higher for smaller boxes. Is this because of density?

Page 13, line 530-1 – the fact that the illusion is “cognitively impenetrable” is not new, I think at least one citation here for this previous finding is necessary.

Typos:

Page 2 line 51: should be “(WTP)”

Page 7 line 241: should be “experiment’s”

Page 7 line 242: should be “whole;” or “whole.” instead of “whole,”

Page 8 line 315: should be “participant’s”

Page 11 line 450: should be “overall”

Page 13 line 509: should be “were” instead of “was”

Reviewer #2: This is a well-written paper interval-based judgements of weight with point judgements of weight. But, while there is much to like about the paper, I feel that is providing a solution to a problem that does not exist. No one in the literature uses fixed point judgements for weight in the way that the authors compare their new method to, so any comparison is simply a straw man. I’m sorry I cannot be more positive, but the study is simply too shallow to warrant publication in this outlet in my view.

The financial disclosures statement isn’t really a statement

No ethical review and approval is rather surprising for this kind of study – presumably the journal has a policy on this?

Line 127 - Order of tasks – where did this factor come from? Surely the task itself is the factor here (and within subject?)

Line 133 – how were the boxes weighted? Was the centre of mass in the physical centre?

Line 133- More information needed on the procedure of lifting each box – it’s quite vague

Point judgement task is quite an odd one – without anything to calibrate to, reporting weight in KG is not something people can do naturally because it’s not something they ever have to do. They can make relative judgements well, but this task sits awkwardly between a scaled judgement and an absolute magnitude estimation. The authors do acknowledge this in the results section, and discuss participant strategies for completing this task.

Line 193 – I think it was three lifts per box, but this is a bit confusingly presented

Line 528 – the line beginning “Several participants showed signs…” isn’t quite worded correctly. But I’m not really sure that those sentences about the well-established cognitive impenetrability of the size-weight illusion are trying to accomplish

This article seems of relevance: https://www.sciencedirect.com/science/article/abs/pii/S0001691814000663

Reviewer #3: The authors measured and compared weight estimates of lifting and hefting objects using two different approaches: “point” and “self-selected interval” judgements – the former being the more prevalent approach. Task objects reflected the size-weight illusion, where objects of the same weight differing in size were presented (smaller ones typically feel much heavier). The authors considered whether the magnitude and variability of the illusion differed depending on which approach was used. I think the procedures were carried our rigorously and hence I won’t comment much about the details. I do have a few larger issues. The paper has merit for publication and it would be great if they can be addressed.

1) I am unable to tell you how much an object weighs in grams. We tried this a bit in my lab and it seems that the ‘psychology’ participants we recruit aren’t good either but, in all fairness, we never examined this systematically. Nonetheless, I am a little surprised that your participants can do this and I would like to know more why and how. What kind of students did you recruit? How did you train them to do this? There was some mention of having lifted a reference mug and how a participant said they were comparing the task objects to a litre of milk which weighs 1 kg (I did not ever think about this until now nor would most people know this). More information is needed. It would also be good to cite more work that examined how well people can judge weights in grams.

2) The two judgement procedures differ in more than one way. Hence, it is difficult to know exactly how the two can lead to differential effects. One concern I have is that they did not match in the number of iterations provided. Namely, for a particular ‘trial’, participants provided one judgment for the “point” approach and multiple ones for the “self-selected interval”. Can resulting differences not because of the strategic process entailed but rather the number of times people could provide an estimate? Would it have been better to allow participants to provide the same number of estimates in the “point” approach?

3) I felt the paper ended anti-climatically. The aim was to compare two methodological approaches and it would be a fair expectation for the reader to receive recommendations as to which of the two approaches should be used and under what contexts.

4) The discussion could also expand on what the results could tell us about the size-weight illusion.

5) I did not understand what you were trying to say in lines 34 to 40. Please unpack.

6) Fix “size-size weight illusion” in line 502.

6. PLOS authors have the option to publish the peer review history of their article (what does this mean?). If published, this will include your full peer review and any attached files.

Reviewer #1: No

Reviewer #2: No

Reviewer #3: No

---

## [Author Response · Author response to Decision Letter 0]

12 Nov 2021

The following responses are included in the rebuttal letter included in this submission. 

Dear Dr. Howe

We have considered the issues and problems raised in the reviews. Please, find below our responses. Our responses are marked with E (editor), R1, R2, and R3 (reviewer 1, 2, and 3).

While you will need to address all the concerns of the reviewers, the following concerns seem to be particularly important:

1) As the first reviewer points out, Buckingham et al. (2014) also had participants make point estimates of the weight of objects, so should be cited.

E.1

 We thank the reviewer for this reference. We added the reference and some text linking our study to that study.

2) Please discuss to what extent density influences weight judgments.

E.2

 We acknowledge that density has been considered to a great extent in the size-weight illusion, SWI literature. We have now added figures describing the relationship between density and accuracy judgments as supplementary material, because density was not the main focus of this paper.

3) As pointed out by the third reviewer, the literature review is incomplete. Please cite and discuss more work that examines the size-weight illusion. To address the concern of the second reviewer, this review should show that the issue you address is not a straw man and that numerical point estimates are common.

E.3

 We added to the references, 15. Buckingham G, Byrne CM, Paciocco J, Eimeren L van, Goodale MA. Weightlifting exercise and the size-weight illusion. Atten Percept Psychophys. 2014 Dec;76(2):452–9., showing that objective weights have been asked for in previous SWI judgements. We also expanded the introductory section about SWI to include some of the different ways used to investigate the illusion.

4) Please describe your participants in more detail – they may be unusually good at this task.

E.4

We had, and still have, no specific reason to believe that our participants were better or worse than the average population within the same age range. Therefore, we did not record demographic data other than age and gender. Furthermore, our main purpose was to compare methods within individuals and we believe that these comparisons should be informative independent of variations in participants performance. 

5) I agree with the third reviewer that there is a potential confound between the conditions – participants provided more estimates in one condition than in the other. This issue needs to be discussed.

 We agree, and have expanded the text in the discussion to cover this possible confound and have suggested ways to deal with this problem in future experiments. 

6) Please indicate which approach should be used in which circumstances.

E.6

 We have expanded the discussion section to include factors that may affect when the interval method is preferred over the over point estimates, for example, that the method may become more useful for very difficult tasks, or with expertise. 

Reviewer #1: 

General comments:

The study is not flashy but I appreciate that it reports a scientifically sound experiment and that these simple experiments that explore methodological issues are of value to the scientific community. I think the paper should be publishable but that some changes are required to increase clarity (see below re issues with the explanations of analyses and also figures) and also substance of the paper.

Specific issues:

1) Page 3, line 96. I believe this paper has assessed the SWI in terms of estimates of weight: Buckingham, G., Byrne, C. M., Paciocco, J., van Eimeren, L., & Goodale, M. A. (2014). Weightlifting exercise and the size–weight illusion. Attention, Perception, & Psychophysics, 76(2), 452-459.

I don’t think this invalidates the current study but it does reduce the novelty. Personally I think the study still has value but the authors need to adjust their statements throughout the manuscript with respect to this issue.

R1.S1

 Thank you for pointing out that we missed this study. We have cited the study and clarified that we only replicate the findings of SWI for judgments of true weight. 

2) The meta-analysis that the authors cite (along with some other papers) strongly suggest that density influences weight judgments including the SWI. The authors report stimulus density but then do not report whether the manipulations impact the effect of density of weight judgments (overall accuracy as well as the SWI). I think this would be a useful addition to the paper and will make it more substantial and informative.

R1.S2

 As mentioned in the earlier response, we have added plots with density and some comments.

Minor issues:

1) Page 1, lines 34-40. “Another benefit in using this approach… in respect to both accuracy and illusion.” I do not understand what the authors are saying here. I think they need to unpack it more.

R1.M1

We have rewritten this paragraph (now starting at line 52) to make it clearer.

2) Page 2 lines 68-80. The relevance of this paragraph for the current study should be made clearer.

R1.M2 

The paragraph (now starting at line 52) has been rewritten to clarify that we described a previous approach studying judgments of intervals and that the main purpose of self-selected intervals should not be affected by participants’ confidence in their judgments, which was found for judgments of intervals. 

3) I think the authors need to explain why they included the average log estimates (Fig 2 right panel).

R1.M3

We have clarified that the responses take a linear form when log transformed in conjunction with the figure and when explaining the multi-level model. 

4) I find Figure 3 to be very unclear and think it needs to be revised. The lines and colours are difficult to see. Maybe use different colours for small/medium vs medium/large or else plot them separately. Also, I think it will be easier to understand if the x axis has the actual weights instead of weight rank (because weight rank is not at all intuitive). Likewise I think similar changes should be made to Figure 4 for the sake of readability.

R1.M4 

 We agree that reporting the ranks obscures the meaning of what we want to communicate. Therefore, figures 3 and 4 were revised so that there are separate panes for small/medium and medium/large comparisons and we hope that they are now more easily interpreted. The x-axis is now marked with weights in grams instead of ranks. The text was also changed to report weights instead of ranks. 

5) Page 9 – I do not understand why the authors describe ANOVAs for the SWI-factor but then do not report them. This needs to be reported or at least explained more clearly.

R1.M5 

 We have clarified that p-values from the F-test should not be interpreted in an exploratory analysis and we only report CI as a measure of confidence/accuracy of our estimates, lines 410 – 414.

6) Page 10, line 395 – what test was done to determine statistical nonsignficance? It should be reported.

R1.M6

 It was dependent t-tests that showed no significance between average accuracy for the different measures, and we have clarified this in the text (line 451).

7) Page 10, final paragraph – now I see that the log weights are analyses but again I do not know why. The authors need to explain this, as well as the maximum likelihood estimation analysis. Also, they refer to Figure 3 here – is this correct?

R1.M7

 We have clarified that the logarithms give a linear response pattern, and that restricted maximum likelihood gives a less biased estimators of variance components according to Zuur et al. (2009). The figure reference has now been corrected to Fig 6 along with table reference also corrected to S8 Table, thank you for pointing this out.

8) Page 11, line 424-5 – hit rates were higher for smaller boxes. Is this because of density?

R1.M8

 As illustrated by the supplementary figure on density, on average, participants assumed different densities for the different sizes of boxes (the point where the regression line fitted to density derived from judgments intersects the true density). For this reason, it is difficult to arrive at the conclusion that density in itself is a determinant of judgment. Furthermore, there are several other variables that may explain why the interval hit-rate was higher. For example, participants used a wider interval for the small box, which increases the chances of including the true weight. The reasons for using wide intervals may vary and we do not have sufficient data, and/or a clear hypothesis, to disentangle this question. 

9) Page 13, line 530-1 – the fact that the illusion is “cognitively impenetrable” is not new, I think at least one citation here for this previous finding is necessary.

R1.M9

 We have added a reference (27. Freeman CG, Saccone EJ, Chouinard PA. Low-level sensory processes play a more crucial role than high-level cognitive ones in the size-weight illusion.) indicating that this has previously been investigated and that our findings are in line with this previous finding. Lines 578-579.

10) Typos:

Page 2 line 51: should be “(WTP)”

Page 7 line 241: should be “experiment’s”

Page 7 line 242: should be “whole;” or “whole.” instead of “whole,”

Page 8 line 315: should be “participant’s”

Page 11 line 450: should be “overall”

Page 13 line 509: should be “were” instead of “was”

R1.M10

 Thank you for noticing and informing us about these typos. The text has been corrected according to your suggestions. 

Reviewer #2: 

General comments: 

This is a well-written paper interval-based judgements of weight with point judgements of weight. But, while there is much to like about the paper, I feel that is providing a solution to a problem that does not exist. No one in the literature uses fixed point judgements for weight in the way that the authors compare their new method to, so any comparison is simply a straw man. I’m sorry I cannot be more positive, but the study is simply too shallow to warrant publication in this outlet in my view.

R2.GC

 Although uncommon in the literature, reviewer 1 pointed out to us that there actually exist at least one other study that has investigated SWI where they collected judgments in the form of kg or pounds. This reference has been added to the manuscript. 

 In everyday life when a person lifts an object and reflects over what it may weigh their thoughts will most probably be in the form of the weight metric they are familiar with. This applies when weight is communicated from one person to another as well. Hence, we used this measure to compare the accuracy in judging this metric for the point and interval judgment methods. 

 All participants thought of a reference weight, and this is now mentioned in the manuscript. Furthermore, many guess the weight competitions can easily be found with for example google indicating that this is something people do outside the lab. 

1) The financial disclosures statement isn’t really a statement

R2.1

 To the best of our knowledge, we have followed the instruction guidelines on what information to provide. In other words, unfortunately we do not know what specific information that should be added outside of what has already been stated to make it a real statement. 

2) No ethical review and approval is rather surprising for this kind of study – presumably the journal has a policy on this?

R2.2

 We have stated to the journal that this type of study does not need approval under current Swedish law. We have also provided with this revision a statement that this is true signed by the deputy head of the department. 

According to Swedish laws, ethics approval is needed only if an intervention is made aimed at changing a person’s physical or mental state. Our method was to only measure their responses to stimuli under the physical and psychological conditions that they arrived to the experiment in, therefore, approval from the Swedish ethics approval authority should not be needed. 

3) Line 127 - Order of tasks – where did this factor come from? Surely the task itself is the factor here (and within subject?)

R2.3

 This paragraph was unclear. The factor was within subject. We have rewritten the text to clarify that we randomized, for each participant (lines 262 – 265), which of the two judgment methods (point or interval) that was performed first. This was used primarily to control for confounds from one task being performed before the other, and no such confounds where found throughout the process of analyzing the data.

4) Line 133 – how were the boxes weighted? Was the centre of mass in the physical centre?

R2.4

 The weights where adhered alongside the inside of the box to achieve a feeling of a uniform weight. The weight was then adjusted by adding cotton inside the box so that precision at the level of single grams could be achieved. A photo illustrating the inside of the boxes has been added (Fig 2) to the manuscript along with text describing the construction. 

5) Line 133- More information needed on the procedure of lifting each box – it’s quite vague

R2.5

 We have clarified (lines 239 – 246) the procedure regarding the one hand grip and how the experiment leader illustrated the grips. 

6) Point judgement task is quite an odd one – without anything to calibrate to, reporting weight in KG is not something people can do naturally because it’s not something they ever have to do. They can make relative judgements well, but this task sits awkwardly between a scaled judgement and an absolute magnitude estimation. The authors do acknowledge this in the results section, and discuss participant strategies for completing this task.

R2.6

 We agree that it may be a bit odd in traditional experimental settings. However, it does reflect how people usually communicate weight to one another as well as being the way weight is usually reported when it is measured by a scale (or other weight measuring methods). 

7) Line 193 – I think it was three lifts per box, but this is a bit confusingly presented

R2.7

 We have clarified that they judged each of the boxes once per session, and that they judged the boxes in each sessions sequential order presented to the participants (lines 247 – 254). 

8) Line 528 – the line beginning “Several participants showed signs…” isn’t quite worded correctly. But I’m not really sure that those sentences about the well-established cognitive impenetrability of the size-weight illusion are trying to accomplish

R2.8

 We have improved the wording of the sentence. We have also rewritten parts of this paragraph (lines 620 – 625) to be clearer about the reasoning about how the stimuli and processes involved may affect whether or not the interval-method is an improvement over traditional point estimates.

9) This article seems of relevance: https://www.sciencedirect.com/science/article/abs/pii/S0001691814000663

R2.9 

 We agree, and it is now included as a reference in the introductory section describing SWI in relation to our experiment. 

Reviewer #3: 

General comments: 

The authors measured and compared weight estimates of lifting and hefting objects using two different approaches: “point” and “self-selected interval” judgements – the former being the more prevalent approach. Task objects reflected the size-weight illusion, where objects of the same weight differing in size were presented (smaller ones typically feel much heavier). The authors considered whether the magnitude and variability of the illusion differed depending on which approach was used. I think the procedures were carried our rigorously and hence I won’t comment much about the details. I do have a few larger issues. The paper has merit for publication and it would be great if they can be addressed.

1) I am unable to tell you how much an object weighs in grams. We tried this a bit in my lab and it seems that the ‘psychology’ participants we recruit aren’t good either but, in all fairness, we never examined this systematically. Nonetheless, I am a little surprised that your participants can do this and I would like to know more why and how. What kind of students did you recruit? How did you train them to do this? There was some mention of having lifted a reference mug and how a participant said they were comparing the task objects to a litre of milk which weighs 1 kg (I did not ever think about this until now nor would most people know this). More information is needed. It would also be good to cite more work that examined how well people can judge weights in grams.

R3.1

 In Sweden where the experiment took place, grams are the most common metric to report weight. Telling the true weight however is hard, and most participants where far from correct. We had no prior expectation that judging weights accurately would be improved by academic education as it is a sensi-motory task. For that reason, we did not keep record of the educational level of our participants and cannot from our data conclude any correspondence between education or ability to judge weights accurately. We did, however, ask participants after the lifting procedure was done if they thought about other weights they knew the weights for, and a liter of milk was a common response. However, these questions where open ended, and used to gather information for development of further studies where these questions can be standardized so that they measure reference weights or other judgment tactics accurately so that it can be compared to judgment accuracy. However, this is outside the scope of the present paper. 

In our open-ended questions at the end of the experiment all participants stated that they used some kind of references weight with 1kg being the most common and 1 liter of milk the most common object representing that 1kg reference weight. However, these responses where too unstructured to be used for an analysis of their effects on the judgments. We have added, to the discussion, information about this and that we encourage future studies to create structured responses to such questions to be able and determine if they are important for whether or not the interval method will be useful for the person. 

2) The two judgement procedures differ in more than one way. Hence, it is difficult to know exactly how the two can lead to differential effects. One concern I have is that they did not match in the number of iterations provided. Namely, for a particular ‘trial’, participants provided one judgment for the “point” approach and multiple ones for the “self-selected interval”. Can resulting differences not because of the strategic process entailed but rather the number of times people could provide an estimate? Would it have been better to allow participants to provide the same number of estimates in the “point” approach?

R3.2

 We agree that number of judgments per item can be a confounding variable. We have added this to the discussion about the time a box is held, and suggested how the two factors can be investigated in future studies to control for this problem. 

3) I felt the paper ended anti-climatically. The aim was to compare two methodological approaches and it would be a fair expectation for the reader to receive recommendations as to which of the two approaches should be used and under what contexts.

R3.3 (same as E.6)

 Unfortunately, we interpret the results as being not clear and/or strong enough to base recommendations or guidelines on. We have provided suggestions for future research to achieve this, and we would like to argue that more research on, for example, what individuals that may or may not benefit from the interval approach, is needed before any concrete advice should be formulated. Such studies can be focused either on judgment strategies, or individual differences in terms of background variables such experiences with lifting weights where the weight in grams are known (e.g. in the gym or in the post-office). 

4) The discussion could also expand on what the results could tell us about the size-weight illusion.

 This question is very interesting and we have expanded further on what the method may reveal about the SWI, at the end of the second paragraph of the discussion section. 

5) I did not understand what you were trying to say in lines 34 to 40. Please unpack.

R3.5

We have rewritten this paragraph (starting at line 52) in an attempt to be more precise and clear.

6) Fix “size-size weight illusion” in line 502.

R3.6

Thank you for noticing, this has been corrected.

Journal Requirements:

and

JR.2 

We have included the title page in the main manuscript file 

3. You indicated that ethical approval was not necessary for your study. Could you please provide further details on why your study is exempt from the need for approval and confirmation from your institutional review board or research ethics committee (e.g., in the form of a letter or email correspondence) that ethics review was not necessary for this study? Please include a copy of the correspondence as an ""Other"" file.

JR.3 

 As we have stated for reviewer 2, comment 2, Approval is needed only if an intervention is made aimed at changing a person’s physical or mental state. Our method was to only measure their responses to stimuli under the physical and psychological conditions that they arrived to the experiment in; therefore, approval from the Swedish ethics approval authority should not be needed. We have also provided with this revision a statement that this is true signed by the deputy head of the department. 

JR.4

No changes are needed. DOIs will be provided if the manuscript is accepted. 

5. Please ensure that you refer to Figure 6 in your text as, if accepted, production will need this reference to link the reader to the figure.

We have corrected the figure numbers throughout the text.

---

## [Decision Letter · Decision Letter 1]

15 Dec 2021

PONE-D-21-18240R1Self-selected interval judgments compared to point judgments: A weight judgment experiment in the presence of the size-weight illusion.PLOS ONE

Dear Dr. Gonzalez,

Thank you for submitting your revised manuscript to PLOS ONE. After careful consideration, we feel that although you addressed most of the concerns raised by the reviewers, there are still some outstanding concerns that you need to address. In particular, before we can accept your manuscript, you need to address the concerns raised by the first reviewer. The other two reviewers are satisfied with the manuscript as it currently stands

We look forward to receiving your revised manuscript.

Kind regards,

Piers D. L. Howe

Academic Editor

PLOS ONE

Journal Requirements:

Reviewers' comments:

Reviewer's Responses to Questions

**Comments to the Author**

1. If the authors have adequately addressed your comments raised in a previous round of review and you feel that this manuscript is now acceptable for publication, you may indicate that here to bypass the “Comments to the Author” section, enter your conflict of interest statement in the “Confidential to Editor” section, and submit your "Accept" recommendation.

Reviewer #1: (No Response)

Reviewer #2: All comments have been addressed

Reviewer #3: All comments have been addressed

2. Is the manuscript technically sound, and do the data support the conclusions?

Reviewer #1: Partly

Reviewer #2: Yes

Reviewer #3: Yes

3. Has the statistical analysis been performed appropriately and rigorously? 

Reviewer #1: No

Reviewer #2: Yes

Reviewer #3: Yes

4. Have the authors made all data underlying the findings in their manuscript fully available?

Reviewer #1: Yes

Reviewer #2: Yes

Reviewer #3: Yes

5. Is the manuscript presented in an intelligible fashion and written in standard English?

Reviewer #1: Yes

Reviewer #2: Yes

Reviewer #3: Yes

6. Review Comments to the Author

Reviewer #1: I think this is an improved, more substantive version of this paper and I appreciate the authors’ work on the manuscript. However, the authors make some claims based on results/data that they have not tested statistically. This is problematic. There are also some other more minor issues that I believe still need to be addressed before this work is publishable.

Major issues:

Line 405 states that there was an ANOVA (actually 2) but then the results are not reported. This is confusing and I still don’t understand why, even they talk about a priori hypotheses etc. In their study aims paragraph in the introduction they state “We also wanted to compare the strength of the illusion across the two different judgment methods”. Then, on line 421 they say “We interpret this as the SWI-reducing effect of the interval method…” as well as similar claims/interpretations in the Discussion. The authors should not claim an effect of a manipulation without having inferential statistical proof of the effect. This needs to be addressed in some way. Similarly, the figure 4 and 5 captions also refer to “interaction plots” which I think is misleading given that they haven’t tested an interaction.

The same issue remains regarding variability (ie SD) in judgements on line 537 onwards. They claim differences that have not been tested statistically.

Lines 620 onwards – this paragraph doesn’t make sense. Not “thinking the illusion away” is what many previous authors have meant by stating it is “cognitively impenetrable”. This is also not what is meant in the literature base by the bottom up vs top down issue. I think the authors needs to research these issues more carefully if they want to include discussion about them.

Also, one of the primary “bottom-up” explanations is related to density perception, which the authors have now referred to in a supplementary figure following my request, but this is still not mentioned in the main manuscript. I still think this is problematic for the reasons I stated in my original review, including that it is reported as an (important) stimulus characteristic. I believe at least a brief discussion of the potential contribution of density to their results and a statement pointing the reader to the supplementary figure is warranted.

Minor issues:

Line 33 “the smaller of two identical (w.r.t shape and size) objects” does not make sense ie there can’t be a smaller of two objects with identical size. Maybe the authors mean weight instead of size?

Line 131 onwards: “Furthermore, focusing on weight, rather than size, has been found to increase the illusion (19). This suggests that asking for the true weight should increase the illusion, because it leads the participants to focus on weight. Hence, differences between judgment methods should be easier to find with a strong illusion when the true weight is judged, instead of subjective heaviness being judged.”

I do not agree with the statement starting with “this suggests…”. I think focusing on true weight instead of heaviness is very different from focusing on weight instead of size. My intuition is that the former would decrease the illusion if anything. I’m not suggesting the authors change their prediction on this issue to accord with mine necessarily but I think whatever they predict on the issue needs to be better substantiated than it is currently. Otherwise I’d say it’s best to remain agnostic on the issue if they cannot.

Typo line 168 – “where” instead of “were”. Same on line 197 “boxes where filled with cotton”.

I think Fig1 was missing.

Fig5 – not sure if it is a problem with the figure or just how it uploaded on the system but the line for mid4 is not visible.

Reviewer #2: The authors have made extensive revisions, for which they should be commended. I still am not overly convinced by the rebuttal to my main concern (the straw man) - the article which is suggested by R1 seems to be done in a very specific context (weight lifters judging the weight of dumbbells, of which presumably they have some expertise already) and the argument that 'guess the weight contests exist' is precisely the argument against why anyone would ever use a point estimate in a weight-judgement task such as this one - because people are pretty hopeless at the task. However I note that the other reviewers didn't share my concerns and, as the manuscript is an otherwise fine piece of work, I do not feel it is my role to block its publication.

Reviewer #3: (No Response)

7. PLOS authors have the option to publish the peer review history of their article (what does this mean?). If published, this will include your full peer review and any attached files.

Reviewer #1: No

Reviewer #2: No

Reviewer #3: No

---

## [Author Response · Author response to Decision Letter 1]

10 Feb 2022

Dear Dr. Howe

We have revised our manuscript in response to the issues raised by reviwer 1. Please, find below our responses. We have numbered the questions 1 – 9 and marked our responses wit R and the corresponding number. Reviewer issues are in italics and responses in normal text. 

Reviewer #1: I think this is an improved, more substantive version of this paper and I appreciate the authors’ work on the manuscript. However, the authors make some claims based on results/data that they have not tested statistically. This is problematic. There are also some other more minor issues that I believe still need to be addressed before this work is publishable.

Major issues:

1. Line 405 states that there was an ANOVA (actually 2) but then the results are not reported. This is confusing and I still don’t understand why, even they talk about a priori hypotheses etc. In their study aims paragraph in the introduction they state “We also wanted to compare the strength of the illusion across the two different judgment methods”. Then, on line 421 they say “We interpret this as the SWI-reducing effect of the interval method…” as well as similar claims/interpretations in the Discussion. The authors should not claim an effect of a manipulation without having inferential statistical proof of the effect. This needs to be addressed in some way. Similarly, the figure 4 and 5 captions also refer to “interaction plots” which I think is misleading given that they haven’t tested an interaction.

R1. We agree that the section was confusing, and we have rewritten it so that it is clearer that it was specifically the peak that we had no prior hypothesis. We have also added ANOVA-analyses of the main effect of measure and the interaction effect with weight causing the peak for point measures. [lines 406 – 437]

2. The same issue remains regarding variability (ie SD) in judgements on line 537 onwards. They claim differences that have not been tested statistically.

R2. We have added t-tests for these differences. [lines 541 – 542]

3. Lines 620 onwards – this paragraph doesn’t make sense. Not “thinking the illusion away” is what many previous authors have meant by stating it is “cognitively impenetrable”. This is also not what is meant in the literature base by the bottom up vs top down issue. I think the authors needs to research these issues more carefully if they want to include discussion about them.

R3. The intended meaning of this paragraph does not add anything of importance and we have therefore removed it.

4. Also, one of the primary “bottom-up” explanations is related to density perception, which the authors have now referred to in a supplementary figure following my request, but this is still not mentioned in the main manuscript. I still think this is problematic for the reasons I stated in my original review, including that it is reported as an (important) stimulus characteristic. I believe at least a brief discussion of the potential contribution of density to their results and a statement pointing the reader to the supplementary figure is warranted.

R4. We have clarified in the stimuli description that density was used to center the entire stimulus set around a parameter and that it was not intended as a main research topic for this study [lines 175 – 178]. We acknowledge that density matters for SWI studies and it is now discussed briefly along with directions to the figure [lines 612 – 619].

Minor issues:

5. Line 33 “the smaller of two identical (w.r.t shape and size) objects” does not make sense ie there can’t be a smaller of two objects with identical size. Maybe the authors mean weight instead of size?

R5. Thank you for pointing out this clearly confusing sentence, it has been rewritten.

6. Line 131 onwards: “Furthermore, focusing on weight, rather than size, has been found to increase the illusion (19). This suggests that asking for the true weight should increase the illusion, because it leads the participants to focus on weight. Hence, differences between judgment methods should be easier to find with a strong illusion when the true weight is judged, instead of subjective heaviness being judged.”

I do not agree with the statement starting with “this suggests…”. I think focusing on true weight instead of heaviness is very different from focusing on weight instead of size. My intuition is that the former would decrease the illusion if anything. I’m not suggesting the authors change their prediction on this issue to accord with mine necessarily but I think whatever they predict on the issue needs to be better substantiated than it is currently. Otherwise I’d say it’s best to remain agnostic on the issue if they cannot.

R6. We agree that the reasoning does not hold when assuming that judgments of true weight should increase the illusion more than subjective judgments of heaviness do. The text has been revised accordingly [lines 132 - 134].

7. Typo line 168 – “where” instead of “were”. Same on line 197 “boxes where filled with cotton”.

R7. We have corrected this, thank you for noticing. 

8. I think Fig1 was missing.

R8. We have checked that the figure should now be included. 

9. Fig5 – not sure if it is a problem with the figure or just how it uploaded on the system but the line for mid4 is not visible.

R9. We have now checked the figures with the Preflight Analysis and Conversion Engine (PACE) as requested by PLOS ONE where no problems are reported and the lines are visible.

---

## [Editor Report · Decision Letter 2]

18 Feb 2022

Self-selected interval judgments compared to point judgments: A weight judgment experiment in the presence of the size-weight illusion.

PONE-D-21-18240R2

Dear Dr. Gonzalez,

We’re pleased to inform you that your manuscript has been judged scientifically suitable for publication and will be formally accepted for publication once it meets all outstanding technical requirements.

Kind regards,

Piers D. L. Howe

Academic Editor

PLOS ONE
---

## [Editor Report · Acceptance letter]

8 Mar 2022

PONE-D-21-18240R2 

Self-selected interval judgments compared to point judgments: A weight judgment experiment in the presence of the size-weight illusion. 

Dear Dr. Gonzalez:

I'm pleased to inform you that your manuscript has been deemed suitable for publication in PLOS ONE. Congratulations! Your manuscript is now with our production department. 

Kind regards, 

on behalf of

Dr. Piers D. L. Howe 

Academic Editor

PLOS ONE